# Comparative Analysis of Surgical Outcomes of Flexible Ureteroscopy and Da Vinci Robotic Surgery in Community Patients with Renal Pelvic Stones Larger than 2 cm

**DOI:** 10.3390/medicina59081395

**Published:** 2023-07-29

**Authors:** Yu-Ju Yeh, Shu-Chuan Weng, Yu-Hsiang Lin, Chien-Lun Chen, Shu-Han Tsao, Han-Yu Tsai, Horng-Heng Juang, Phei-Lang Chang, Chen-Pang Hou

**Affiliations:** 1Department of Urology, Chang Gung Memorial Hospital at Linkou, Taoyuan 333, Taiwan; yalulu86725@gmail.com (Y.-J.Y.); linyh@cgmh.org.tw (Y.-H.L.); clc2679@cgmh.org.tw (C.-L.C.); b9702054@cgmh.org.tw (S.-H.T.); b9802087@cgmh.org.tw (H.-Y.T.); henryc@cgmh.org.tw (P.-L.C.); 2Department of Health and Management, Yuanpei University of Medical Technology, Hsinchu 330, Taiwan; scweng303@mail.ypu.edu.tw; 3Bachelor Degree Program of Senior Health and Management, Yuanpei University of Medical Technology, Hsinchu 330, Taiwan; 4School of Medicine, Chang Gung University, Taoyuan 333, Taiwan; hhj143@cgu.edu.tw

**Keywords:** stones, RIRS, RAPL, robotic, ureteroscopy

## Abstract

*Background and Objectives*: This study evaluated and compared the surgical outcomes of retrograde intrarenal surgery (RIRS) lithotripsy versus robot-assisted laparoscopic pyelolithotomy (RAPL) in community patients with renal pelvic stones larger than 2 cm. *Materials and Methods*: A total of 77 patients who underwent RIRS (RIRS group, *n* = 50) or RAPL (RAPL group, *n* = 27) at our institution between December 2016 and July 2022 were recruited. A single surgeon performed all surgical operations. Preoperative, operative, and postoperative data were recorded. The study evaluated various clinical outcomes, namely, urinary tract infections, analgesic use, emergency room readmissions, stone clearance rates, surgical complications, and medical expenditures associated with the treatment courses, and compared them between the groups. *Results:* The RAPL group had a larger mean stone diameter and higher degree of hydronephrosis than the RIRS group did. The RIRS group had superior outcomes regarding operative time, length of postoperative hospital stay, surgical wound pain, and medical expenditures. Regarding postoperative outcomes, comparable rates of postoperative urinary tract infection, prolonged analgesic use, and emergency room readmissions were observed between the groups. However, the RAPL group had a higher stone clearance rate than the RIRS group did (81.5% vs. 52.0%, *p* = 0.014). *Conclusions:* For the surgical treatment of renal pelvis stones larger than 2 cm, RAPL has a superior stone clearance rate than RIRS; however, RIRS achieves superior outcomes in terms of medical expenditures, length of hospital stay, and surgical wound pain. Both procedures were equally safe.

## 1. Introduction

Urolithiasis is a social and economic concern worldwide, and an increasing number of individuals are affected by the condition [1]. The prevalence of urolithiasis ranges from 1% to 13% worldwide [2]. In Taiwan, the age-adjusted prevalence of upper urinary tract stone disease was approximately 7.3%, with a higher likelihood of occurrence in men compared to women [3]. Common symptoms of urolithiasis include severe pain in the back, lower abdomen, and groin, along with dysuria, hematuria, nausea, and vomiting [4]. Urolithiasis is influenced by various risk factors, such as high body mass index, insufficient fluid intake, genetic predisposition, socioeconomic status, and environmental conditions. [5,6,7,8]. The treatment approach for renal stones varies depending on the specific characteristics of the stone, the overall health of the patient, and other individual factors. Treatment options include extracorporeal shock wave lithotripsy, ureteroscopy, percutaneous nephrolithotomy (PCNL), retrograde intrarenal surgery (RIRS), and open surgery [9]. According to the guidelines, PCNL is recommended as the primary option for renal stones larger than 2 cm. However, RIRS can serve as an alternative when PCNL is contraindicated, even for stones exceeding 2 cm in size [9]. PCNL is considered a risky procedure, with an overall complication rate as high as 30.3%. One of the major concerns associated with PCNL is postoperative bleeding [10]. Other potential complications include renal parenchymal injury (15.4%), perioperative bleeding (6.3%), delayed bleeding (0.9%), and renal collecting duct injury (5.2%) [10]. Therefore, for kidney stones larger than 2 cm, alternative, minimally invasive procedures have recently been proposed, including RIRS and robot-assisted laparoscopic pyelolithotomy (RAPL); these procedures do not cause direct injury to the parenchyma of the kidneys, unlike PCNL [11,12]. RIRS is an effective treatment for renal stones larger than 2 cm. RIRS offers several benefits over PCNL, such as reduced risk of bleeding and shorter hospital stays [12]. Although PCNL and RAPL are safe and efficient methods of managing large renal stones, they are associated with a significantly higher mean estimated blood loss when compared with PCNL; furthermore, the stone-free rate is lower [13]. Despite increasing acceptance of these parenchymal injury-free surgical methods among urologists in recent years, few studies have directly compared these methods. Accordingly, the present study analyzed and compared the clinical prognosis of these operations.

## 2. Materials and Methods

### 2.1. Patient Selection

Patients who underwent RIRS or RAPL at our institution between December 2016 and July 2022 were recruited. Prior to surgery, all patients underwent evaluation and screening by the responsible surgeon to determine their suitability for minimally invasive surgery. This study received approval from the Institutional Review Board of Chang Gung Memorial Hospital in Taiwan. Before receiving treatment, the patients were informed of the process, advantages, disadvantages, and risks associated with the treatment options and signed consent forms. Adult patients were included if they had a single large (≥2 cm) unilateral renal pelvis stone and if they were suitable candidates for RIRS or RAPL. All included patients had received medical treatment only and had not undergone any invasive procedures, such as ureteric stenting or nephrostomy, prior to inclusion. Patients who had an Eastern Cooperative Oncology Group performance status of ≥2 [14], an active urinary tract infection, multiple scattered stones, a stone tamponade occupying more than two renal calyx or concomitant ureter stones, a history of urinary organ cancer or congenital abnormalities, or previously undergone urinary system reconstruction surgery were excluded. To maintain objectivity, the surgical procedures were conducted by a single surgeon. After receiving a comprehensive explanation of the surgical process, the patients were given the freedom to choose whether to undergo RIRS or RAPL. The sizes of stones, determined according to the longest diameter, were assessed using computed tomography, intravenous pyelography, or plain radiography of the kidney, ureter, and bladder (KUB). The degree of hydronephrosis was evaluated through computed tomography or ultrasonography according to the hydronephrosis grading system [15].

### 2.2. Surgical Procedures and Techniques

RAPL was performed using a fourth-generation da Vinci Xi Surgical System (Intuitive Surgical, Sunnyvale, CA, USA). The surgical procedures and techniques, including patient placement, port placement, colon mobilization, renal pelvis dissection, extended pyelolithotomy, and stone removal, followed the methods described by Madi et al. [16]. Diagnostic semirigid ureteroscopy was performed on patients undergoing RIRS to assess their ureters. During the procedure, a ClearPetra access sheath (Micro-Tech Endoscopy, Ann Arbor, MI, USA) was placed through a guide wire under fluoroscopy guidance. Retrograde access to the upper urinary tract was achieved using a flexible fiberoptic ureteroscope (Olympus America, MN, USA). Once the calculus was located, a 200-µm laser fiber was inserted and stones were fragmented using a 30-W Holmium:YAG laser generator (SphinxX; Lisa, Katlenburg-Lindau, Germany). In both groups of patients, in accordance with standard practice, 5–6-Fr 26–28 cm double-J stents (Polaris, Boston Scientific, MA, USA) were inserted at the end of the procedure to prevent ureteral obstruction or infection.

### 2.3. Postoperative Care and Follow-Up

The standard protocol for antibiotic administration involved administering a single dose of preventive cefazolin intravenously, followed by administering oral cefadroxil (500 mg) twice daily for 7 days. The choice of antibiotics was determined according to urine culture results prior to surgery. Both procedures followed a standard pain management plan, which included taking acetaminophen for 7 days. Follow-up observations were conducted using kidney, ureter, and bladder (KUB) radiography or renal ultrasound 1 day, 1 week, and 1 month after surgery. Stone clearance was defined as the absence of visible residual stones or the presence of stone fragments no larger than 3 mm in the postoperative image taken 1 month after the first surgery. The study evaluated various clinical outcomes, including urinary tract infections, analgesic use, emergency room readmissions, stone clearance rates, surgical complications, and medical expenses associated with the treatment courses, and compared them between the groups. Medical expenditure encompassed expenses recorded in the medical invoicing system of our institution and included fees related to admission, primary procedures, salvage treatments, and all follow-up examinations conducted within 1 month following the operation.

### 2.4. Statistical Analysis

The relationships between categorical variables were examined using a chi-square test, and mean values were compared between the groups by using an independent-samples *t*-test. The likelihood of an outcome occurring in one group compared with the other group was assessed using relative risk values. A *p* value of <0.05 was considered to indicate statistical significance for all statistical analyses. Data were analyzed in SPSS (version 25; IBM: Armonk, NY, USA).

## 3. Results

A total of 77 patients were enrolled. Among these patients, 50 underwent RIRS and 27 underwent RAPL. Patient assessments were performed preoperatively and postoperatively, and follow-up lasted at least 6 months. No significant differences in age, gender, American Society of Anesthesiologists physical status classification [17], surgical history, or comorbidities were observed between the two groups (Table 1). Patients who underwent RAPL had larger average stone diameters than those who underwent RIRS did (34.3 vs. 23.3 mm, *p* = 0.001). Additionally, the incidence and intensity of hydronephrosis were significantly greater in the RAPL group than in the RIRS group (*p* = 0.001). Data on past history of stone management surgery and selected treatment options are presented in Table 2. Among the patients who underwent RIRS, 42.0% had no history of stone management surgery; however, 55.6% of the patients who underwent RAPL had no history of stone management surgery. Extracorporeal shock wave lithotripsy was the most frequently employed interventional treatment prior to the surgical procedures in the present study. Table 3 provides information about the utilization of different imaging examinations both before and after the procedures for the assessment and follow-up of patients. Before the operation, the patients underwent different imaging examinations according to the needs of their conditions. However, most patients underwent KUB and renal ultrasound as follow-up tools after the operation.

The perioperative variables in each group are listed in Table 4. Operative time was greater in the RAPL group than in the RIRS group (174.7 vs. 77.2 min, *p* < 0.001). Additionally, the length of postoperative hospital stay was longer in the RAPL group than in the RIRS group (3.6 vs. 1.6 days, *p* < 0.001). No instances of intraoperative complications were observed in either group. Postoperative pain was evaluated using a numeric rating scale (NRS) [18]. On the first day after surgery, the proportions of patients with NRS scores of ≤1 were 76.0% and 22.2% in the RAPL and RIRS groups, respectively (*p* < 0.001). Similarly, on the second day after surgery, the proportions of patients with NRS scores of ≤1 were 88.0% and 37.0% in the RAPL and RIRS groups, respectively (*p* < 0.001). These findings indicate that patients in the RIRS group experienced significantly lower levels of postoperative pain than those in the RAPL group did. One patient in the RIRS group remained in hospital for more than 5 days because he developed postoperative acute pyelonephritis and required intravenous antibiotics. Two patients in the RAPL group remained in hospital for 7 and 17 days separately. One of these patients experienced recurrent urinary retention, necessitating a wait for spontaneous urination before being discharged. The other patient, weighing approximately 110 kg, experienced acute renal failure associated with rhabdomyolysis, which was possibly triggered by muscle compression due to prolonged immobility. This patient was transferred to the nephrology ward for treatment until his condition stabilized, after which he was discharged. The mean medical cost in the RAPL group was NT$193,153, significantly exceeding that in the RIRS group (NT$82,013, *p* < 0.001).

Postoperative outcomes in each group are listed in Table 5. No significant differences were observed between the groups in terms of the development of a urinary tract infection within 1 month or dependence on oral analgesics for more than 1 week after surgery. Moreover, no significant differences were observed between the groups in terms of the rate of emergency department admission within 1 month after surgery. Two patients in the RIRS group and one patient in the RAPL group returned due to acute pyelonephritis. One patient in the RAPL group had urinoma resulting from poor healing of the renal pelvis suture. Candidiasis was confirmed in this patient after urine culture. Computed tomography–guided pigtail drainage and antifungal treatment successfully resolved the infection. Stone clearance was defined as the absence of visible residual stones or the presence of stones no larger than 3 mm in imaging examinations conducted 1 month after surgery. On the basis of this definition, the RAPL group had a significantly higher stone clearance rate than the RIRS group did (81.5% vs. 52.0%, *p* = 0.014). This indicates that RAPL is more effective in treating stones larger than 2 cm. The reintervention rate was 24% in the RIRS group and 18.5% in the RAPL group, with no significant difference between the groups. Extracorporeal shock wave lithotripsy was the reintervention method in both groups.

## 4. Discussion

According to the European Association of Urology guidelines on urolithiasis, both extracorporeal shock wave lithotripsy and RIRS are equally viable treatment options for renal stones that are smaller than 2 cm in diameter; however, for stones exceeding 2 cm in diameter, PCNL is recommended [19]. RIRS is not recommended for stones exceeding 2 cm in diameter because it achieves worse stone clearance outcomes than does PCNL, engendering the need for staged procedures for effective treatment [20]. Another study revealed that although RIRS has proven effective in treating renal stones of 2–4 cm, it is crucial to acknowledge that multiple procedures may be necessary to address the entire stone burden completely [21]. A recent meta-analysis demonstrated that RIRS is a safe and efficient procedure for specific patients with large renal stones; nevertheless, PCNL was demonstrated to have a higher postoperative stone clearance rate than RIRS in patients with renal stones larger than 2 cm did [22]. Although PCNL is a well-established treatment for renal stones larger than 2 cm, it has several drawbacks. Specifically, PCNL is considered a high-risk procedure and has an overall complication rate of 30.3% [10]. One of the major concerns associated with PCNL is postoperative bleeding [10]. Other potential complications include renal parenchymal injury, perioperative bleeding (6.3%), delayed bleeding (0.9%), and renal collecting duct injury (5.2%) [10]. Therefore, for kidney stones larger than 2 cm, alternative minimally invasive procedures have been proposed, including RIRS and RAPL; these procedures do not cause direct injury to the renal parenchyma, are considered safe, and achieve high stone clearance rates [23]. Another meta-analysis demonstrated the safety of robotic surgery for treating renal stones. Robotic surgery causes less blood loss and achieves higher stone clearance rates than PCNL does [13]. Interest among urologists in surgical methods that minimize parenchymal injury has increased. However, few studies have directly compared RIRS with RAPL. Accordingly, the present study comprehensively analyzed the clinical outcomes of these two procedures and compared them between two groups.

Our study revealed no differences in age, gender, or health status between the two groups. The choice of surgical approach was made according to stone dimensions and the presence of hydronephrosis. Due to the absence of a standard reference, our study was conducted to address the issue of how surgeons select between these two procedures. Our investigation revealed that stone size and hydronephrosis played significant roles in shared decision making. Surgeons and patients tended to opt for RAPL over RIRS when dealing with large stones or stones within the renal pelvis causing hydronephrosis. Extraction of stones from an enlarged renal pelvis is simpler with a robotic arm. Furthermore, an extended incision for large stones, particularly irregularly shaped stones, in the renal pelvis, is more conveniently achieved using a robotic arm. Since our study was not randomized, there was a notable disparity in stone size between the RIRS and RAPL groups. Consequently, comparing the operation times will inevitably yield inequalities. Large stones, for instance, may raise concerns about potential infection when undergoing RIRS. Additionally, fragmented stones caused by RIRS may migrate into the ureter, leading to obstruction. On the contrary, smaller stones may raise concerns regarding cost-effectiveness if the robotic arm is utilized for the surgical procedure. Both RIRS and RAPL are safe and effective treatments for kidney stones larger than 2 cm. Neither surgical approach resulted in intraoperative complications in our study. However, RIRS outperformed RAPL in various aspects, such as operative time, length of hospital stay, postoperative pain, and overall medical expenses. Among the 77 patients in the present study, 7 (9.1%) experienced perioperative complications, of whom 5 experienced minor complications (Clavien–Dindo score [24] I–II) and 2 experienced severe complications (Clavien–Dindo score IIIa). The two patients with severe complications were in the RAPL group. Neither surgical method resulted in severe hematuria, bleeding-related complications, or events requiring blood transfusions due to low hemoglobin levels. A comprehensive retrospective study, including the largest PCNL cohort in the literature, observed that the most prevalent surgical complications were postoperative fever and bleeding-related complications, including instances of septic shock [25]. Nevertheless, none of these complications were observed in the present study. Our findings suggest that both RIRS and RAPL are a safe alternative to traditional PCNL for treating kidney stones larger than 2 cm, provided they are performed without causing any damage to kidney tissue.

A concern with RAPL for kidney stones is the risk of peritonitis or abscess formation due to the infectious nature of urolithiasis. In the present study, RAPL was performed intraperitoneally. This approach provides optimal exposure. When the ureter is incised, urine containing bacteria from the renal pelvis can flow into the abdominal cavity, potentially leading to intra-abdominal infection. No instances of intra-abdominal infection were observed in the present study. In another study, the infection rate associated with intraperitoneal RAPL was lower than anticipated [26]. In the mentioned study, no instances of sepsis or postoperative fever were observed, and only one occurrence of urinary tract infection was noted [26]. Adel et al. [27] reported that among 20 patients who received transperitoneal laparoscopic pyelolithotomy, only 1 experienced peritonitis because of retained stones in the abdominal cavity. Another study focusing on laparoscopic and robotic pyelolithotomy found no instances of postoperative peritonitis or infection [28]. The aforementioned studies, including the present study, concluded that the intraperitoneal approach did not elevate the risk of intra-abdominal infection. This is likely due to the presence of bacteria commonly found in infectious stones, such as *Escherichia coli*, *Proteus mirabilis*, *Klebsiella pneumoniae*, and *Staphylococcus aureus*, which were the predominant species detected in urine and abscess bacterial cultures. Notably, *P. mirabilis*, a prevalent pathogen responsible for renal stone infection, is uncommon among patients with peritonitis [29]. Previous studies have shown that even without robotic surgery, it is feasible to use traditional laparoscopy to achieve favorable surgical outcomes for renal stones. Siforoosh et al. conducted a study involving 28 patients who underwent Laparoscopic Pyelolithotomy (LPL) for the treatment of Large Renal Stones with Intrarenal Pelvis Anatomy [30]. Out of these patients, three experienced urinary leak, and one required double J insertion [30]. Another parallel-group randomized clinical trial demonstrated that, compared to PCNL, LPL yielded a higher stone-free rate and reduced bleeding in patients with single or limited particles staghorn stones with extrarenal pelvis, albeit with a longer operation duration [31]. Additionally, a separate study revealed that LPL is a safe and effective method for managing large kidney stones in pediatric patients [32]. The Korean Society of Endourology and Robotics (KSER) also recommended that a laparoscopic approach, either conventional or robotic assisted, may be advantageous if there are concomitant anatomic abnormalities, such as ureteropelvic junction stricture, ureteral stricture, or renal diverticulum [33].

A slightly lower stone clearance rate was observed in the present study compared with those reported in the literature [22,23]. This discrepancy might be because we adopted a relatively strict definition of stone clearance: the absence of visible residual stones or the presence of fragments measuring no more than 3 mm in diameter in postoperative images taken 1 month after the initial surgical session. Nevertheless, patients with residual stones who underwent further extracorporeal shock wave lithotripsy achieved complete stone clearance during the 6-month follow-up period. Additionally, our study revealed that despite the RAPL group having larger average stone diameters prior to surgery compared with the RIRS group, the group’s stone clearance rate was higher.

This study has several limitations. First, the number of cases in our study was too small to use propensity score matching (PSM) to select our cases. Patient allocation was determined through shared decision-making [34] rather than randomization and the data analysis was conducted retrospectively. Because this was not a randomized case–control trial, potential bias may be present that could affect the objectivity of the findings. Second, the medical expenditure in this study included costs documented in the hospital invoicing system charged within 1 month after surgery. However, several patients sought treatment for postoperative issues at other clinics or hospitals, and these expenses could not be included in the calculation. Third, the follow-up period was relatively short, averaging only 6 months. According to Traxer and Thomas [35], the use of a ureter access sheath during RIRS resulted in a 46.5% incidence of ureteral wall injuries. Owing to the limited follow-up period, we could not assess late complications, such as ureteral strictures, which can manifest weeks to years after surgery [36]. Finally, due to the retrospective nature of this study, significant clinical information was absent, including the stone density observed in the images and the post-surgery quality of life questionnaire completed by the patients. Despite these limitations, we maintain that this study is innovative and holds substantial clinical significance. To the best of our knowledge, this is presently the sole comprehensive comparative study evaluating the clinical outcomes of RIRS and RAPL. Our findings provide confirmation that both RIRS and RAPL, when executed with minimal parenchymal injury for treating kidney stones exceeding 2 cm in diameter, present a safe alternative to conventional PCNL.

## 5. Conclusions

RIRS and RAPL, when conducted as procedures with minimal parenchymal injury for treating kidney stones larger than 2 cm, can be a viable and safe alternative to traditional PCNL. However, each surgical method has its own set of advantages and disadvantages. Compared with RIRS, RAPL achieves superior stone clearance rates but results in greater surgical wound pain, higher medical expenses, and longer hospital stays.

## Figures and Tables

**Table 1 medicina-59-01395-t001:** Pre-operative data of the patients.

	RIRS (*n* = 50)	RAPL (*n* = 27)	*p* Value
Age (range, SD)	56.5 (33–81, 10.2)	56.2 (36–79, 11.5)	0.910
Male/Female	32/50	21/6	0.303
Stone diameter (mm) (range, SD)	22.3 (20–35, 5.30)	34.3 (20–100, 16.7)	0.001
ASA score ≥ III, n(%)	22 (44.0%)	11 (40.7%)	0.783
Stone intervention naïve	21 (42.0%)	15 (55.6%)	0.255
Comorbidity factors	DM: 14 (28.0%)	DM 7 (25.9%)	0.845
	HTN: 28 (56.0%)	HTN: 11 (40.7%)	0.201
	CAD: 2 (4.0%)	CAD: 0 (0%)	0.539
	Stroke: 5 (10.0%)	Stroke: 1 (3.7%)	0.417
	Cr > 1.3 ng/dL: 3 (6.0%)	Cr > 1.3 ng/dL: 8 (29.6%)	0.013
Hydronephrosis	Grade 0: 34 (68.0%)	Grade 0: 7 (25.9%)	0.001
	Grade 1–2: 7 (14.0%)	Grade 1–2: 5 (18.5%)	
	Grade 3–4:9 (18.0%)	Grade 3–4: 15 (55.6%)	

Abbreviations: RIRS: retrograde intrarenal surgery; RAPL: robotic assisted pyelolithotomy; SD: standard deviation; ASA: The American Society of Anesthesiologists; DM: diabetes mellitus; HTN: hypertension; CAD: coronary arterial disease; Cr: creatinine.

**Table 2 medicina-59-01395-t002:** Previous stone management history and present treatment.

	RIRS, *n* (%)	RAPL, *n* (%)	*p* Value
Intervention naive	21 (42.0)	15 (55.6)	0.599
ESWL	20 (40.0)	8 (29.6)	
URSL	4 (8.0)	3 (11.1)	
Open or PCNL	4 (8.0)	1 (3.7)	

Abbreviations: RIRS: retrograde intrarenal surgery; RAPL: robotic assisted pyelolithotomy; ESWL: extracorporeal shockwave lithotripsy; URSL: ureterorenal scope lithotripsy; PCNL: percutaneous lithotripsy.

**Table 3 medicina-59-01395-t003:** Image examinations for pre-operative evaluation and post-operative follow-up.

	RIRS (*n*, %)	RAPL (*n*, %)
Pre-operative evaluation		
KUB	50 (100)	27 (100)
Renal ultrasound	30 (60)	18 (66.7)
IVP	11 (22)	6 (22.2)
CT	27(54)	20 (74.1)
Post-operative follows up		
KUB	50 (100)	27 (100)
Renal ultrasound	50 (100)	27 (100)
IVP	1 (2)	0
CT	3 (6)	2 (7.4)

Abbreviations: RIRS: retrograde intrarenal surgery; RAPL: robotic assisted pyelolithotomy; IVP: Intravenous Pyelogram; KUB: Kidneys, Ureters, and Bladder; CT: Computed Tomography.

**Table 4 medicina-59-01395-t004:** Perioperative data.

	RIRS	RAPL	RIRS vs. RAPL	
			RR (95% CI)	*p* Value
OP time (minutes) (range, SD)	77.2 (39–214, 41.85)	174.7 (44–387, 77.52)		<0.001
Postop hospital stay (days) (range, SD)	1.6 (1–5, 0.93)	3.6 (1–17, 2.94)		<0.001
Intra op complications	0	0		
NRS ≤ 1 (day 1)	38 (76.0%)	6 (22.2%)	0.42 (0.264–0.671)	<0.001
NRS ≤ 1 (day 2)	44 (88.0%)	10 (37.0%)	0.32 (0.159–0.644)	<0.001
hospital stay > 5 days	1 (2%) APN	2 (7.4%) Urinary retention: 1 Rhabdomyolysis with acute renal injury: 1	1.98 (0.398–9.924)	0.28
Medical expenditure (NTD)	82,012.8 (72,132–118,810, 13,598)	193,153 (152,932–266,429, 28,333)		<0.001

Abbreviations: RIRS: retrograde intrarenal surgery; RAPL: robotic assisted pyelolithotomy; RR: relative risks; OP: operation; SD: standard deviation; NRS: numeric rating scale; APN: acute pyelonephritis; NTD: new Taiwan dollar.

**Table 5 medicina-59-01395-t005:** Postoperative data.

	RIRS	RAPL	RIRS vs. RAPL	
			RR (95% CI)	*p* Value
UTI (<1 month)	10 (20%)	3 (11.1%)	0.813 (0.571–1.157)	0.32
Analgesics require (>1 week)	5 (10%)	2 (7.4%)	0.9 (0.546–1.484)	1
Returned ER (<1 month)	2 (4%)	2 (7.4%)	1.315 (0.487–3.553)	0.609
Hydronephrosis	2 (4%)	2 (7.4%)	1.315 (0.487–3.553)	0.609
Stone clearance	26 (52.0%)	22 (81.5%)	2.900 (1.249–6.735)	0.004
Re-intervention	12 (24%)	5 (18.5%)	0.802 (0.358–1.800)	0.775

RIRS: retrograde intrarenal surgery; RAPL: robotic assisted pyelolithotomy; OP: operation; RR: relative risks; CI: confidence interval; UTI: urinary tract infection; ER: emergent room.

## Data Availability

The data used to support the findings of this study are available from the corresponding author upon request.

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
