# Peer review of "Comparative Analysis of Surgical Outcomes of Flexible Ureteroscopy and Da Vinci Robotic Surgery in Community Patients with Renal Pelvic Stones Larger than 2 cm"

_medicina, 2023, doi:10.3390/medicina59081395_

Round 1
Reviewer 1 Report
This is a Comparative analysis of surgical outcomes of flexible ureteoscopy and Da Vinci robotic surgery in community patients with renal pelvic stones larger than 2 cm. The paper is clear and well written. Stone clearance was in favour of RAPL Major concerns: Sample size low, no PSM could be applied to the cohort. No randomization. Authors should provide stone density of the patients in table 1. Strenghts: single surgeon prospective enrollement Lastly, QoL questionaires could enforce this paper. Please add this as a limitation if they are not available. Example DOI: 10.23736/s2724-6051.23.04882-6. PMID: 37067185.
Author Response
Response to Reviewer 1 Comments
Dear Reviwer:
We appreciate your valuable suggestions. We have addressed your inquiries and made some revisions to our manuscript based on your feedback. We look forward to your reply and the agreement to have our manuscript considered for acceptance.
Thank you.
Points 1 :
Major concerns: Sample size low, no PSM could be applied to the cohort. No randomization. Authors should provide stone density of the patients in table 1. Strenghts: single surgeon prospective enrollement. Lastly, QoL questionaires could enforce this paper. Please add this as a limitation if they are not available.
Response:
We appreciate your suggestion. We agree with your viewpoint that the number of cases included in our study needs to be increased to utilize propensity score matching (PSM) to select our cases. Instead of randomization, patient allocation was determined through shared decision-making. As this study did not follow a randomized case-control trial design, potential bias may exist, which could impact the objectivity of the findings. Furthermore, since this is a retrospective study, we did not document the stone density of images and the post-surgery quality of life for the patients. We recognize these limitations in our research and have included the limitations you raised in the revised version. (Line 320, 332, type in red). We genuinely appreciate your recommendations and aim to address these issues in future studies.
Reviewer 2 Report
The authors compared the preformace of RIRS vs. robotic pyelolithotomy (RAPL) in patients with renal stones > 2cm. The following items need consideration:
MAJOR
As stated by the authors, the choice between RIRs or RAPL has not been randomized and based on shared dicision making by the surgeon-patient. However, the authors are required to fully explain the criteria on which the single surgeon chose between the two methods. The authors should also indicate whether in the time frame of the study 2016-2022, both procedures were done or if a procedure was introduced later or discontinued earlier.
A MAJOR concern with the results of this study is the GREAT difference in the size of operated stones between the two study groups which makes these assumed comparison groups totally different. The largest size of stone in the RIRS group is 35mm (Table 1) while the average size of stone in the RAPL group is around 35 mm. The largest size of stone in group RAPL is 100 mm. Therefore, I assume that the bulk of stone is usually a second power of estimates of stone diameter should be about 2.25 times the bulk in group RIRS. These great differences call into question the initial comparability of groups in this study. In my opinion as the largest stone size in group RIRS is equal to the average stone size in group RAPL then the longest operation duration in group RIRS was better to be compared with the mean operation duration in group RAPL. In my opinion as differences between study groups are great regarding stone size which is one of the most important factors in relation to performance and complications in RIRS and RAPL, even multivariable analysis cannot be relied on to control for this confounding effect.
MINOR
Introduction lines 34 to 57 can be extensively summarized as they are not directly related to the scope of this research.
Lines 69-71 there is a mistake in typing.
I do not agree with the authors that the risk of infection in PCNL is more than RIRS. In fact PCNL is an operation done is low pressure field while RRIS is done in a higher pressure conditions. Therefore, risk of backflow into renal prenchyma and pyelonephritis is higher is RIRS for equal size of stones.
Line 72: RIRS and RAPL are not injury-free methods, they do not injure parenchyma but have their own profile of injury.
Please indicate whether in the RIRS group, there has been occasions in which RIRS could not be done in the first attempt and a ureteral catheter was inserted and patients were refereed for a later session of RIRS: how many if yes.
preoperative imaging: how many patients were imaged by KUB or CT or IVP of a combination of methods?
postoperative imaging: how many patients were imaged by KUB or CT or IVP of a combination of methods?
The authors are encouraged to present the location of stone in kidney in tables (pelvis, upper, middle , lower calyx, or staghorn).
The authors are encouraged to present the presence of single versus multiple stone particles in kidneys in their tables.
Did the author use RAPL for kidneys with intrarenal pelvis anatomy ? please explain.
DISCUSSION
As some centers do not have robot, I encourage authors to include laparoscopic pyelolithotomy in line with robotic pyelolithotomy in their discussion. The following references are suggested as there are other references in the literature from which the authors may choose:
Simforoosh N, Radfar MH, Valipour R, Dadpour M, Kashi AH. Laparoscopic Pyelolithotomy for the Management of Large Renal Stones with Intrarenal Pelvis Anatomy. Urol J. 2020 Apr 13;18(1):40-44. doi: 10.22037/uj.v0i0.5576. PMID: 32281090.
Soltani MH, Hossein Kashi A, Farshid S, Mantegy SJ, Valizadeh R. Transperitoneal Laparoscopic Pyelolithotomy versus Percutaneous Nephrolithotomy for Treating the Patients with Staghorn Kidney Stones: A Randomized Clinical Trial. Urol J. 2021 Dec 20;19(1):28-33. doi: 10.22037/uj.v18i.6831. PMID: 34927230.
Erçil H, Karkin K, VuruÅŸkan E. Is laparoscopic pyelolithotomy an alternative to percutaneous nephrolithotomy for treatment of kidney stones larger than 2.5 cm in pediatric patients? Pediatr Surg Int. 2023 Jan 11;39(1):78. doi: 10.1007/s00383-023-05367-4. PMID: 36627447.
needs minor editing only.
Author Response
Response to Reviewer 2 Comments
Dear Reviewer:
We appreciate your valuable suggestions. We have addressed your inquiries and made some revisions to our manuscript based on your feedback. We look forward to your reply and the agreement to have our manuscript considered for acceptance.
Thank you.
Point 1 :
As stated by the authors, the choice between RIRS or RAPL has not been randomized and based on shared dicision making by the surgeon-patient. However, the authors are required to fully explain the criteria on which the single surgeon chose between the two methods. The authors should also indicate whether in the time frame of the study 2016-2022, both procedures were done or if a procedure was introduced later or discontinued earlier.
Response :
The surgical procedures for these patients took place from 2016 to 2022, and the surgical approach selection was based on a shared decision-making process. Both of these methods were simultaneously employed during the same time frame, without any specific order or sequence. Hence, due to the absence of a standard reference, our study was conducted to address how surgeons select between these two procedures. Our investigation revealed that two factors play a role in the surgeon's decision-making process: the stone size and hydronephrosis. Both surgeons and patients exhibited less preference for RIRS when dealing with larger stones. Conversely, surgeons were more inclined to opt for RAPL when stones were located in the renal pelvis, causing hydronephrosis. Using a robotic arm simplifies the extraction of stones from an enlarged renal pelvis. Additionally, a robotic arm offers greater convenience in performing an extended incision for larger stones, especially irregularly shaped ones in the renal pelvis. Consequently, we have incorporated the abovementioned findings into our revised discussion section (line 247, type in red)
Point 2:
In my opinion as the largest stone size in group RIRS is equal to the average stone size in group RAPL then the longest operation duration in group RIRS was better to be compared with the mean operation duration in group RAPL. In my opinion as differences between study groups are great regarding stone size which is one of the most important factors in relation to performance and complications in RIRS and RAPL, even multivariable analysis cannot be relied on to control for this confounding effect.
Response:
We fully concur with your observation. Since our study was not randomized, there was a notable disparity in stone size between the RIRS and RAPL groups. Consequently, comparing the operation times will inevitably yield inequalities. Nevertheless, we discovered that stone size undeniably impacts the decision-making process for both physicians and patients. Large stones, for instance, may raise concerns about potential infection when undergoing RIRS. Additionally, fragmented stones caused by RIRS may migrate into the ureter, leading to obstruction. On the contrary, smaller stones may raise concerns regarding cost-effectiveness if the robotic arm is utilized for the surgical procedure. We sincerely appreciate you raising this point, and we have incorporated the information mentioned above into our study's revised version. ( Line 254, type in red)
Point 3:
Introduction lines 34 to 57 can be extensively summarized as they are not directly related to the scope of this research.
Response :
We have made the above sentence more concise. ( Line 34-42, type in red)
Point 4:
Lines 69-71 there is a mistake in typing.
I do not agree with the authors that the risk of infection in PCNL is more than RIRS. In fact, PCNL is an operation done is low pressure field while RRIS is done in a higher pressure conditions. Therefore, risk of backflow into renal prenchyma and pyelonephritis is higher is RIRS for equal size of stones.
Response:
Thank you for correcting our mistake. We reviewed the cited literature. The author's research content is as follows: The hospital stays (weighted mean difference [WMD] = -2.10, 95% CI -3.08 to -1.11, p < 0.10) and bleeding (RR = 0.20, 95% CI 0.06-0.68, p = 0.01) were lower and operation time was longer (WMD = 19.11, 95% CI 7.83-30.39, p < 0.10) for RIRS. Indeed, no mention was made of a lower Infection rate. This is obviously our mistake; we have fixed it in the revised version. (Line 56, type in red)
Point 5:
Line 72: RIRS and RAPL are not injury-free methods, they do not injure parenchyma but have their own profile of injury.
Response:
We concur with your viewpoint that RIRS and RAPL are free of renal parenchymal injury instead of injury-free approaches. In the revised version, we have made the necessary amendments accordingly. ( Line 60, type in red )
Point 6:
Please indicate whether in the RIRS group, there has been occasions in which RIRS could not be done in the first attempt and a ureteral catheter was inserted and patients were refereed for a later session of RIRS: how many if yes.
Response:
Most of the circumstances you mentioned primarily occurred in cases involving ureteral stones. In situations of chronic inflammation, ureteral stenosis, or significant ureteral angulation resulting from hydronephrosis, inserting a double-J catheter and proceeding with staged RIRS is advisable. However, our research focuses specifically on renal pelvis stones, and fortunately, we did not encounter the circumstances mentioned above in our study.
Point 7:
Preoperative imaging: how many patients were imaged by KUB or CT or IVP of a combination of methods?
Postoperative imaging: how many patients were imaged by KUB or CT or IVP of a combination of methods?
Response:
We have followed your suggestion, and in our updated version, we have separately described the preoperative and postoperative imaging in Table 3
Point 8:
The authors are encouraged to present the location of stone in kidney in tables (pelvis, upper, middle, lower calyx, or staghorn).
The authors are encouraged to present the presence of single versus multiple stone particles in kidneys in their tables.
Response:
Our study specifically selected cases with a single renal pelvis stone based on the preoperative imaging examination. This decision was made because the effectiveness of stone clearance is influenced not only by the stone size but also by factors such as the number of stones and the location of the calyx. To ensure the objectivity of our study, we focused exclusively on cases involving single renal pelvis stones. We have made it clearer in our revised version. ( Line 73, type in red)
Point 9:
Did the author use RAPL for kidneys with intrarenal pelvis anatomy? please explain.
Response:
After re-checking the preoperative imaging of our patients, we discovered that only two patients exhibited Intraparenchymal renal pelvic anatomy, as defined by Tomaszewski et al. in 2023. Both of them underwent RIRS in our study. Intraparenchymal renal pelvic anatomy is a rare anatomical variation that is linked to a higher occurrence of urine leakage after partial nephrectomy. However, our patients who underwent RAPL did not display Intraparenchymal renal pelvic anatomy.
Reference : Tomaszewski JJ, Cung B, Smaldone MC, et al. Renal pelvic anatomy is associated with incidence, grade, and need for intervention for urine leak following partial nephrectomy. Eur Urol. 2014;66(5):949-955. doi:10.1016/j.eururo.2013.10.009
Point 10:
As some centers do not have robot, I encourage authors to include laparoscopic pyelolithotomy in line with robotic pyelolithotomy in their discussion. The following references are suggested as there are other references in the literature from which the authors may choose
Response:
We appreciate your suggestion, and as a result, we have incorporated the studies you mentioned into our reference list. Furthermore, we have incorporated the findings from these studies into our discussion. ( Line 294, type in red )
Reviewer 3 Report
Congratulation on the present study! In front of the developing technologies, renal stone disease management benefits from many options.
The paper is well-designed and written, but I have some observations which are detailed point by point in the following:
1. The introduction may be shortened. There are too many epidemiological data. It should be more focused on surgical procedures.
2. You can include the guideline specifications for surgical procedures in the introduction.
3. Was the project designed according to the Declaration of Helsinki? It should be mentioned.
4. The authors can include a chart or scheme in the Material and Methods section.
5. How did the authors select the patient for one procedure or another? It should be mentioned.
6. Is it possible to include a table to detail the costs per technique used?
7. You can include this study in the discussion section https://doi.org/10.3390/medicina59010124
The English language is fine.
Author Response
Response to Reviewer 3 Comments
Dear Reviewer:
We appreciate your valuable suggestions. We have addressed your inquiries and made some revisions to our manuscript based on your feedback. We look forward to your reply and the agreement to have our manuscript considered for acceptance.
Thank you.
Point 1:
The introduction may be shortened. There are too many epidemiological data. It should be more focused on surgical procedures.
Response:
Based on your suggestion, we have condensed the introduction section, particularly the epidemiology part, to enhance its conciseness. ( Line 34, type in red )
Point 2 :
You can include the guideline specifications for surgical procedures in the introduction.
Response:
According to the quidelines, PCNL is recommended as the primary option for renal stones larger than 2 cm. However, RIRS can serve as an alternative when PCNL is contraindicated, even for stones exceeding 2 cm in size. We've added this to the introduction and provided references. ( Line 44, type in red)
Point 3 :
Was the project designed according to the Declaration of Helsinki? It should be mentioned.
Response:
Our study is in line with the principles of the Declaration of Helsinki. This study is based on a thorough knowledge of the scientific background, a careful assessment of risks and benefits, a reasonable likelihood of benefit to the population studied, and conducted by suitably trained investigators. As mentioned in the material and methods section, this study received approval from the Institutional Review Board of Chang Gung Memorial Hospital in Taiwan. Before receiving treatment, the patients were informed of the process, advantages, disadvantages, and risks associated with the treatment options and signed consent forms.
Point 4 :
The authors can include a chart or scheme in the Material and Methods section.
Response:
This study did not include a chart or scheme as it was purely retrospective, focusing on two distinct surgical approaches. The study design was very simple. The treatment method for the selected patients was determined through a shared-decision making process, and all participants completed the treatment and underwent a follow-up period of at least 6 months.
Point 5:
How did the authors select the patient for one procedure or another? It should be mentioned.
Response:
As stated in our manuscript, patient allocation in our study was determined through a shared decision-making process. Prior to treatment, patients were provided with detailed information regarding the treatment options, including the procedure itself, its advantages, disadvantages, and associated risks. Following a comprehensive explanation of the procedures, patients were given the autonomy to choose between RIRS and RAPL, and consent forms were signed accordingly. Our investigation revealed that stone size and hydronephrosis played significant roles in shared decision-making. Surgeons and patients tended to opt for RAPL over RIRS when dealing with large stones or stones within the renal pelvis causing hydronephrosis. ( Line70, Line 82, Line 244, type in red )
Point 6 :
Is it possible to include a table to detail the costs per technique used?
Response:
Unfortunately, we are unable to provide a comprehensive listing of the table you suggested. This is because there are not only various surgical techniques available, but each patient also requires unique medications, treatment for internal medical conditions, and their length of hospitalization can differ. Additionally, the types of hospital rooms vary. Consequently, compiling an exhaustive list of every detail would be a very complex task. The medical expenses covered in our institution's medical invoicing system consist of fees associated with admission, primary procedures, salvage treatments, and all follow-up examinations conducted within one month after the operation. The hospital's reimbursement system can only obtain the total medical expenses.
Point 7:
You can include this study in the discussion section https://doi.org/10.3390/medicina59010124
Response:
We appreciate your suggestion. According to the study, the author mentioned that RIRS has proven to be effective in treating renal stones larger than 2 cm. However, it is crucial to acknowledge that multiple procedures may be necessary to completely address the entire stone burden. We have added this reference to our manuscript. ( Line 224, type in red)
Round 2
Reviewer 2 Report
The authors have done their best to deal with the raised concerns and done most of them. Nevertheless, some concerns are not rectifiable by the nature of the study as admitted by the authors and reflected in their discussion.
Author Response
Response to the reviewer:
Point 1:
The authors have done their best to deal with the raised concerns and have done most of them. Nevertheless, some problems are not correctable by the nature of the study, as admitted by the authors and reflected in their discussion.
Response:
Dear Reviewer:
Thank you so much for your guidance and comprehension regarding our circumstances. This study is retrospective, and we have thus far obtained only initial research findings. Nevertheless, we firmly believe that our research yields valuable insights and can provide clinicians with practical and effective clinical information. Our ultimate aim is to conduct prospective and randomized case-control studies in the future. To achieve this, we must steadily accumulate the number of cases to ensure more objective preliminary outcomes.
Once more, we express our gratitude for your valuable suggestion.
Sincerely
Chen-Pang Hou